# The Spatial Analysis and Enhancement of Social Housing in Seoul

**Sang-Hyun Chung [1] and Jun Ha Kim [2],***

[1] School of Architecture, Sungkyunkwan University, Suwon-si 16419, Republic of Korea; chun0069@skku.edu
[2] Department of Housing & Interior Design, Kyung Hee University, Seoul 02447, Republic of Korea
* Correspondence: junhakim@khu.ac.kr; Tel.: +82-10-7403-7487

**Abstract:** Seoul is the first city in South Korea to provide affordable social housing, beginning in 2015. Despite the importance of studying its space, configuration, and functionality, which impact the residents' quality of life, research on social housing in Seoul is sparse. The purpose of this study is twofold: (1) to analyze the spatial layout and utilization of social housing and (2) to compare it to the layout and utilization of public rental housing provided by Seoul Housing & Communities Corporation (SH). A total of 120 units from 30 social housing projects open to the public in Seoul were selected. This study found that social housing in Seoul primarily consists of compact housing units targeting young adults and newlywed couples, with a high proportion of diverse spatial configurations. In South Korea, while the SH specialized floor plan is commonly used as a standardized prototype in contemporary apartments, the floor plans for social housing exhibit a broader range of shapes and spatial configurations. Distinct criteria are employed for space separation in designing social housing. Separating bedrooms or kitchens may be the priority depending on the specific housing unit, resulting in multiple layouts. The results suggest that ongoing research could contribute to exploring improved approaches for region-specific social housing design and enhancing residential environments.

**Keywords:** social housing; spatial layout; residential environment; Seoul

## 1. Introduction

Although housing is a fundamental human right, securing this right remains a challenge, especially in Seoul, South Korea. The country's rapid economic growth and Seoul's swift urbanization since 1970, with the city's population increasing from 5.6 million to 10 million in 1988, has exacerbated the difficulty in finding adequate housing [1]. While Seoul's population has shown a slight decline since 2013 due to population dispersion policies in the metropolitan area [1], Seoul still has the highest population density (15,551 people/km$^2$) in South Korea [2]. Owning a home in the city still demands considerable financial resources.

The price-to-income ratio (PIR) is an indicator used to measure the time required to purchase an average-sized house in a specific area, considering the annual income [3]. It is a metric for housing affordability [4]. As of 2020, Seoul's PIR stood at 24.01 years [5]. This implies that one needs to save for 24 years without spending a single penny to afford a home. Seoul ranks highest in the PIR among OECD countries, with Paris following at 22.02 years (second), London at 21.21 years (third), Tokyo at 13.97 years, and New York at 10.76 years [6]. According to the PIR based on household income, most major cities have high PIR values, indicating significant costs and time needed for homeownership. Seoul's lead in the PIR hierarchy globally underscores the challenge of homeownership and the scarcity of affordable housing.

According to the 2018 OECD statistics, the ratio of social housing to total housing is remarkably low in South Korea (6.8%) compared to several European countries, such as



the Netherlands (37.7%), Denmark (21.2%), Austria (20.0%), the United Kingdom (16.9%), France (12.7%), and Ireland (12.7%) [6]. These data indicate that compared to many Western countries, South Korea still falls behind in social housing provision.

In particular, for Seoul's young adults and newlyweds, owning a home has become even more difficult due to increased property prices, high unemployment rates, and education loan burdens [7–9]. Social housing has emerged as a realistic alternative since 2015 to address these issues. Currently, the Seoul Metropolitan Government is expanding the supply of social housing for low- to middle-income households, ensuring stable rental periods with prices lower than market rates [7]. Social housing contributed to urban regeneration that gradually expanded through small-scale development rather than supplying through large-scale apartment complexes [10].

Since the operation period of social housing has been relatively short compared to public rental housing in South Korea, most studies have focused on analyzing the policies, support measures, and supply status of foreign countries' social housing. For example, Oh (2017) analyzed the supply policies of social housing in the UK and Germany, and Im (2015) reviewed Germany's social housing system and housing support laws [11,12]. Similarly, Song (2018) examined the core aspects of welfare housing in the Netherlands and its implications, and Kim (2015) explored the developmental process and implications of social housing in the Netherlands [13,14].

However, there is a significant lack of research on the spatial analysis of social housing. European countries, which have implemented social housing policies earlier than South Korea, have gone through trial and error to enhance the spatial functions and utilization of social housing to improve citizens' quality of life. For example, in Spain, there are standards in square meters depending on the number of types of rooms [15]. This underscores the importance of spatial analysis, given that the spatial functions and utilization of social housing directly impact residents' quality of life.

Therefore, this study analyzes the spatial layout and utilization of social housing in Seoul and compares the spatial layout and utilization between social housing in Seoul and public rental provided by the Seoul Housing & Communities Corporation (SH). This study will offer insights into developing effective solutions and enhancements in the spatial design of social housing.

## 2. Social Housing in Seoul

### 2.1. The Concept of Social Housing

Seoul was the first city in South Korea to provide social housing [16]. Interestingly, the private sector took the lead in providing social housing for the first time in 2012 [16]. During this process, Seoul introduced social housing initiatives pioneered by the private sector and enacted the Social Housing Support Ordinance in 2015 [10]. According to this ordinance, social housing is defined as "rental housing supplied by social and economic entities targeting socially and economically vulnerable individuals". Social and economic entities include social enterprises, cooperatives, small ventures, and community businesses. Unlike public rental housing in Korea, which is funded purely by the government [17], social housing in Seoul involves a partnership between these entities and the Seoul Metropolitan Government [13]. The key distinction between social housing and public rental housing lies in their providers; the former involves both public and private sectors, while the latter is purely public.

### 2.2. Eligibility for Social Housing in Seoul

The target beneficiaries for social housing include urban workers whose monthly average income is below 70% of the median income [18]. The rent fee for social housing is set at 80% or below the market rate, and the tenancy period is from 6 to 10 years [13].

### 2.3. Types of Social Housing Projects in Seoul

The social housing projects pursued in Seoul can be broadly divided into two types since enacting the Social Housing Support Ordinance: the land lease type and the remodeling type. The land lease type involves Seoul City, the Real Estate Investment Trust, and the Korea Land and Housing Corporation (LH Corporation, Jinju-si, Republic of Korea) leasing their land to private developers [19]. These developers then construct and operate housing on the leased land. The remodeling type involves remodeling and re-leasing aging or vacant housing units [14]. As of 2020, Seoul had approximately 2000 units of social housing in supply [20].

### 2.4. Distribution of Social Housing in Seoul

Currently, there are over 100 social housing projects listed on the social housing platform operated by Seoul City [19]. They are mostly concentrated in the following areas: Geumcheon-gu (14 projects), Gwanak-gu (11 projects), Eunpyeong-gu (10 projects), Seongbuk-gu (8 projects), Gangbuk-gu (8 projects), and Guro-gu (6 projects).

The social housing distribution in Seoul's districts, along with the apartment values for each administrative district, are shown in Figure 1.

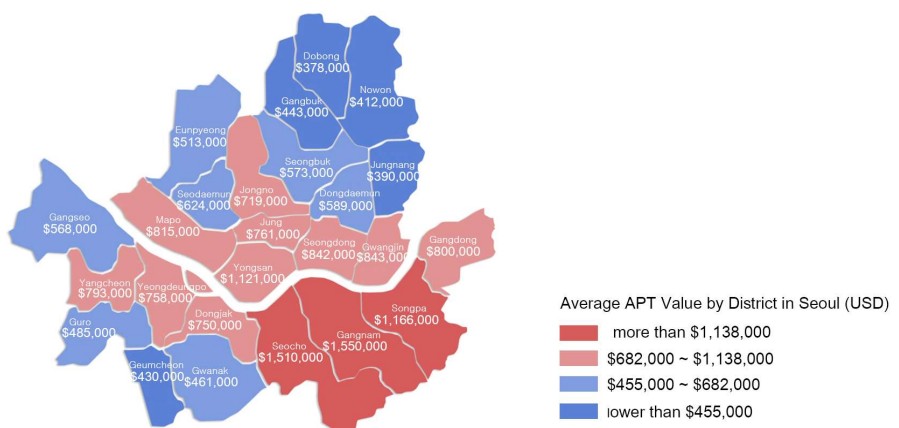

**Figure 1.** Seoul metropolitan area average apartment prices and social housing status by district. (Notes: Data were derived from Real Estate 114 [21], and the authors geographically represented the locations of social housing projects on the map. The numbers on the map indicate the number of units in each social housing project.).

The density of one-person households among young people in Seoul and the distribution of social housing showed a limited correlation. However, meaningful results were observed when examining the relationship with the apartment price distribution by district. Only two or three social housing projects were distributed in areas with the highest apartment prices, such as Gangnam-gu and Seocho-gu. Conversely, in areas with the lowest apartment prices, like Geumcheon-gu, the highest number of social housing projects, 14 in total, were distributed. Additionally, there were 11 projects in Gwanak-gu, 10 projects in Eunpyeong-gu, 8 projects in Seongbuk-gu and Gangbuk-gu, and 6 projects in Guro-gu. Social housing seems more prevalent in areas with lower apartment prices. It is worth noting that while the target residents for social housing are primarily young individuals living alone or newlywed couples, the social housing distribution does not strongly correlate with the preferences of these target groups. Instead, the distribution appears to be significantly influenced by the real estate costs in each area.

### 3. Methodology

*3.1. Data*

A total of 120 units from 30 social housing projects in Seoul were selected based on publicly available layout data. These data came from the Seoul City Social Housing Platform [22].

*3.2. Spatial Analysis*

Spatial analysis involves examining and evaluating spatial patterns and relationships within a given space [10]. Spatial analysis is utilized to scrutinize the layout and utilization of social housing units in this paper. Spatial configuration refers to the arrangement and layout of spaces, specifically within social housing units in Seoul. This analysis includes a study of various room types (e.g., living room, dining room, kitchen, bathroom, terrace), focusing on understanding their spatial characteristics, such as size and arrangement.

First, the characteristics of social housing units were analyzed. Second, the spatial configuration of these units was examined. Third, the spatial configurations of social housing in Seoul were compared with public rental housing. In Seoul, public rental housing was provided by SH, a local public enterprise established under the Seoul Metropolitan Government. The goal of the SH is to stabilize the housing of Seoul citizens and improve their quality of life through public rental housing [23].

### 4. Results

*4.1. Classification of Spatial Types in Seoul Social Housing*

A total of 120 units of 30 social housing projects were selected and analyzed to classify the spatial types of social housing in Seoul. Units were categorized into Studio, 1 Bed, 2 Bed, 3 Bed, or Multiple, based on the number of separate bedrooms. The major spaces within each unit, including the living room, dining room, kitchen, bathroom, and terrace, were identified and classified accordingly.

For the main spaces of each unit, such as the living room, dining room, and kitchen, their degree of spatial separation was denoted with a "+" sign, indicating separate spaces, while spaces without spatial separation were represented without the "+". In this study, the LDK analysis method was employed to categorize and explore the connectivity and partitioning methods of housing floor plans.

The unit floor plan diagrams used abbreviations in English for the major spaces mentioned above to indicate the spatial characteristics of the configuration of these spaces: L (living room), D (dining room), K (kitchen), B (bathroom), T (terrace), etc. In the case of bedrooms, when using the LDK notation, a number was placed before the abbreviation to represent the number of bedrooms. The net floor area of each unit was represented in square meters ($m^2$).

*4.2. Unit Plan Space Configuration Characteristics*

First, as shown in Table 1, among units with an area of 30 $m^2$ or less, Studio had the highest proportion, accounting for 31.7% (38 units), followed by 1 Bed (4.2%), 2 Bed (0.8%), and n Bed (0.0%). Second, among units with an area of 31 $m^2$–40 $m^2$, 1 Bed had the highest percentage, at 11.7%, followed by 2 Bed (0.0%) and n Bed (0.0%). Third, among units with an area of 41 $m^2$–59 $m^2$, Studio accounted for 0%, 1 Bed (12.5%), 2 Bed (10%), and n Bed (2.5%). In the 60 $m^2$–84 $m^2$ and 85 $m^2$ or larger categories, the total proportion was low, at 5.8% and 3.3%, respectively.

**Table 1.** Summary of unit plan space configuration characteristics (n = 120).

| Classification | Category | n (%) |
|---|---|---|
| Net floor area | $30 \, \text{m}^2 \leq$ | 44 (36.7%) |
| | $31–40 \, \text{m}^2$ | 35 (29.2%) |
| | $41–59 \, \text{m}^2$ | 30 (25.0%) |
| | $60–84 \, \text{m}^2$ | 7 (5.8%) |
| | $85 \, \text{m}^2 \geq$ | 4 (3.3%) |
| | Mean (SD): 41.57 (23.78), min: $23.66 \, \text{m}^2$, max: $188.72 \, \text{m}^2$ | |
| Number of bedrooms | Studio | 59 (49.2%) |
| | 1 Bed | 35 (29.2%) |
| | 2 Bed | 14 (11.7%) |
| | n Bed | 12 (10.0%) |
| Space composition | Open | 44 (36.7%) |
| | Separate (Alcove studio, 1 bed, 2 bed, n bed) | 76 (63.3%) |
| LDK configuration | L + D + K | 20 (16.7%) |
| | L + DK | 1 (0.8%) |
| | LD + K | 31 (25.8%) |
| | LDK | 68 (56.7%) |
| Balcony | No | 20 (16.7%) |
| | Multi-purpose | 82 (68.3%) |
| | Yes | 18 (15.0%) |
| Number of units | Mean (SD): 15.18 (7.36), min: 3, max: 40 | |
| Number of floors | 3 < (Single Family House) | 10 (8.3%) |
| | 3–4 (Townhouse) | 5 (4.2%) |
| | 5 ≥ (Apartment) | 105 (87.5%) |

Note: Response percentages may not add up to 100% due to rounding.

Social housing in Seoul, primarily designed for single-person households and newlywed couples, exhibits a concentration of Studio and 1 Bed types, and the highest distribution is found in small-sized homes of $30 \, \text{m}^2$ or less.

Regarding the net floor area, units with $30 \, \text{m}^2$ or less accounted for 36.7%, followed by $31 \, \text{m}^2–40 \, \text{m}^2$ (29.2%), $41 \, \text{m}^2–59 \, \text{m}^2$ (25.0%), $60 \, \text{m}^2–84 \, \text{m}^2$ (5.8%), and $85 \, \text{m}^2$ or more (3.3%).

In terms of housing type, the proportion of units with separated rooms (63.3%) was higher than that of open-plan units (36.7%). The unit plan composition revealed that LDK layouts constituted more than half of the total (56.7%), followed by LD + K (25.8%), L + D + K (16.7%), and L + DK (0.8%) types. As for the balcony type, multipurpose balconies were dominant, accounting for 68.3%, while units without balconies represented 16.7%, and units with dedicated balcony spaces accounted for 15.0%.

Regarding the number of floors, units with five floors or more were the most prevalent, making up 87.5% of the total. Units with three floors or less constituted 8.3%, and units with three to four floors accounted for 4.2%.

It was observed that unit plans with a net floor area of $30 \, \text{m}^2$ or less were the most commonly distributed, with Studio types and open-plan units that did not differentiate between living room, dining room, and kitchen being the most prevalent configurations.

### 4.3. LDK Configuration Types

The combination of unit size characteristics and LDK configuration types yielded the analysis results presented in Table 2. Among units with a net floor area of 30 m$^2$ or less, the LDK configuration was most prevalent in Studio types, accounting for 28.3%. The next most common types were LDK configurations in Studio types within the 31 m$^2$–40 m$^2$ range, which accounted for 9.2%, and LD + K configurations in Studio types within the 31 m$^2$–40 m$^2$ range, which accounted for 6.72%. In the 60 m$^2$–84 m$^2$ and 85 m$^2$ or larger categories, the LDK configuration was represented at 5.8% and 3.3%, respectively, indicating a lower distribution in larger units.

**Table 2.** LDK configuration types of social housing (n = 120).

| Number of Bedrooms | LDK Configuration | Total | 30 m$^2$ $\leq$ | 31–40 m$^2$ | 41–59 m$^2$ | 60–84 m$^2$ | 85 m$^2$ $\geq$ |
|---|---|---|---|---|---|---|---|
| Studio | LDK | 45 (37.5%) | 34 (28.3%) | 11 (9.2%) | 0 (0.0%) | 0 (0.0%) | 0 (0.0%) |
| | LD + K | 12 (10.0%) | 4 (3.3%) | 8 (6.7%) | 0 (0.0%) | 0 (0.0%) | 0 (0.0%) |
| | L + D + K | 2 (1.7%) | 0 (0.0%) | 2 (1.7%) | 0 (0.0%) | 0 (0.0%) | 0 (0.0%) |
| | Sub-total | 59 (49.2%) | 38 (31.7%) | 21 (17.5%) | 0 (0.0%) | 0 (0.0%) | 0 (0.0%) |
| 1 Bed | 1 LDK | 15 (12.5%) | 4 (3.3%) | 6 (5.0%) | 5 (4.2%) | 0 (0.0%) | 0 (0.0%) |
| | 1 LD + K | 12 (10.0%) | 1 (0.8%) | 6 (5.0%) | 5 (4.2%) | 0 (0.0%) | 0 (0.0%) |
| | 1 L + D + K | 8 (6.7%) | 0 (0.0%) | 2 (1.7%) | 5 (4.2%) | 1 (0.8%) | 0 (0.0%) |
| | Sub-total | 35 (29.2%) | 5 (4.2%) | 14 (11.7%) | 15 (12.5%) | 1 (0.8%) | 0 (0.0%) |
| 2 Bed | 2 LDK | 7 (5.8%) | 1 (0.8%) | 0 (0.0%) | 5 (4.2%) | 1 (0.8%) | 0 (0.0%) |
| | 2 LD + K | 5 (4.2%) | 0 (0.0%) | 0 (0.0%) | 5 (4.2%) | 0 (0.0%) | 0 (0.0%) |
| | 2 L + D + K | 2 (1.7%) | 0 (0.0%) | 0 (0.0%) | 2 (1.7%) | 0 (0.0%) | 0 (0.0%) |
| | Sub-total | 14 (11.7%) | 1 (0.8%) | 0 (0.0%) | 12 (10.0%) | 1 (0.8%) | 0 (0.0%) |
| n Bed | nLDK | 1 (0.8%) | 0 (0.0%) | 0 (0.0%) | 0 (0.0%) | 0 (0.0%) | 1 (0.8%) |
| | nLD + K | 3 (2.5%) | 0 (0.0%) | 0 (0.0%) | 2 (1.7%) | 1 (0.8%) | 0 (0.0%) |
| | nL + D + K | 8 (6.7%) | 0 (0.0%) | 0 (0.0%) | 1 (0.8%) | 4 (3.3%) | 3 (2.5%) |
| | Sub-total | 12 (10.0%) | 0 (0.0%) | 0 (0.0%) | 3 (2.5%) | 5 (4.2%) | 4 (3.3%) |
| Total | | 120 (100%) | 44 (36.7%) | 35 (29.2%) | 30 (25.0%) | 7 (5.8%) | 4 (3.3%) |

Note: Response percentages may not add up to 100% due to rounding.

The results show that relatively more unit plans were concentrated in the 30 m$^2$ or less category, accounting for 36.7% of the total, followed by 29.2% in the 31 m$^2$–40 m$^2$ category. This suggests that the unit sizes for Seoul's social housing are mainly focused on 30 m$^2$ or less and 31 m$^2$–40 m$^2$.

Regarding the LDK configuration types, the most prevalent was the open-plan LDK configuration in Studio units, accounting for 37.5%. The next most common was the 1 LDK configuration in Studio units, where the living room, kitchen, and dining room are open plan (12.5%), followed by the LD + K configuration in Studio and 1 Bed units, where the kitchen is separate (10.0%). In Studio types, the L + D + K configuration, where the living room, dining room, and kitchen are separate, accounted for only 1.7%, showing a significantly lower frequency of separate configurations.

Considering that Seoul's social housing targets young adults aged 19 to 39 or newlywed couples, with a focus on young single-person households, it can be observed that most housing types are expected to be small units, mainly Studio open-plan configurations without separate LDKs, except for shared houses when excluding remodeling models. The analysis results revealed that while Studio open-plan configurations constituted 37.5% of all LDK types, the combined proportions of Studio LD + K and Studio L + D + K configura-

tions, as well as 1 Bed and 2 Bed types, amounted to 52.6%, indicating a higher presence of separate configurations.

Analyzing the unit size distribution, it was found that among Studio units, those with an area of 30 m$^2$ or less accounted for the highest percentage at 31.7%. Combining the 30 m$^2$ or less 1 Bed units (4.2%) and 2 Bed units (0.8%) constituted 36.7% of the total. In contrast, Studio units with an area of 31 m$^2$ or larger were 17.5%, 1 Bed units were 25%, and 2 Bed units were 10.8%, making up a total of 53.3% for unit sizes 31 m$^2$ or larger. Comparing the area distribution of Studio, 1 Bed, and 2 Bed units, 36.7% accounted for sizes of 30 m$^2$ or less, while 53.3% were sizes of 31 m$^2$ or larger.

The analysis results suggest that instead of heavily focusing on the smallest unit sizes, social housing in Seoul considered resident convenience and favored configurations with separate studios or 1 Bed and 2 Bed layouts. This differs from the initial prediction that small-scale housing targeting young single-person households would be the dominant type.

### 4.4. Comparison of Social Housing Space Types

From the comparison, we observed that the distribution of unit sizes and spatial configurations is not solely focused on small-sized open types. Consequently, we further examined the actual spatial compositions to discern their distinct characteristics. A comparative analysis was conducted between the spatial configurations of social housing in Seoul and the SH specialized floor plans, emphasizing variations in unit sizes and layouts, particularly focusing on Studio, Alcove Studio, 1 Bed, and 2 Bed configurations.

As shown in Table 3, It is evident that major indoor spaces, such as the living room, dining area, and bedroom, converge into a singular area after examining the interior layout of the Studio type. The SH specialized floor plans, however, are designed in a rectangular fashion, mirroring the typical architecture of apartments or officetels in South Korea, encompassing a total area of 24.96 m$^2$. The size does not differ significantly compared to the Studio type found in social housing.

**Table 3.** Comparison of social housing floor plans and SH specialized floor plans (Studio type).

| SH Specialized Floor Plan | Social Housing Floor Plan | | |
|---|---|---|---|
|  |  |  |  |
| SH_Green Light (24.96 m$^2$) | Together House II (27.14 m$^2$) | Star Hills A (25.65 m$^2$) | Banya Ville (27.40 m$^2$) |

The Banya Ville floor plan follows a layout similar to the SH specialized floor plan, with an entrance that leads directly into a succession of essential areas. Conversely, both the Together House II and the Star Hills A feature a square design, deviating from the rectangular outline of the SH specialized floor plan. These designs place the main space, which doubles as the living room and dining area, centrally, and it can also function as a bedroom. Such a layout is consistently applied across all studio-type floor plans, though the placement of the entrance, bathroom, and kitchen might vary.

Although the unique characteristics of the Studio type make direct comparisons of its merits and demerits a challenge, juxtaposing it with the SH specialized floor plan reveals a more diverse range of spatial layouts in social housing plans. Typically, the Studio type merges living, dining, and sleeping areas into an open environment, with subtle variations in the locations of the entrance, bathroom, and kitchen. The comparative analysis between the Studio type of social housing and the SH specialized floor plan underscores the diversity of spatial configurations present in social housing floor plans.

In the case of the Alcove Studio type, as shown in Table 4, the SH specialized floor plan utilizes two bays on the external side to separate the bedroom and living room, placing the kitchen at the rear for the dining area and living room to share one space. Like the Studio type, the floor plan also considers the possibility of balcony expansion to add 10 m$^2$ of space. In contrast, the Air Sillim III social housing floor plan has a similar layout to the SH specialized floor plan, while the Together House V floor plan shows a completely different form of Alcove Studio type. In the SH specialized floor plan, the bedroom is separated, but in the second example of the Together House V floor plan 5, the kitchen and dining area are located on the left side upon entering the entrance, and the bedroom and living room are shared, while the dining area is separated. In the example of the Together House V floor plan, the entrance is located at the center, and the kitchen, dining area, and living room are integrated into one space, while the bedroom is planned on the other side.

**Table 4.** Comparison of social housing floor plans and SH specialized floor plans (Alcove Studio type).

| SH Specialized Floor Plan | Social Housing Floor Plan | | |
|---|---|---|---|
|  |  |  |  |
| SH_Green Light (29.91 m$^2$) | Star Hills C (26.03 m$^2$) | Air Sillim III (35.16 m$^2$) | Together House V (35.21 m$^2$) |

The Alcove Studio type in social housing can be classified into two types. The first type is similar to the SH floor plan, where the bedroom is separated as a personal space, and the shared spaces, such as the living room and dining area, are planned in one space. The priority of this type is to separate the bedroom. The second type includes configurations where the bedroom and living room are integrated into one space while the kitchen and dining area are separated. In this type, priority is given to separating the kitchen and dining area from the living room. When comparing the SH specialized floor plan and social housing floor plans, they can be broadly divided based on the method of separating the living room from the bedroom or separating the living room from the dining area, catering to the preferences and tastes of the residents.

As shown in Table 5, in the case of the 1 Bed type, the SH specialized floor plan shows an expanded form of the Alcove Studio type. It separates the living room and bedroom from the front two bays and places the kitchen and dining area at the rear entrance. Like other specialized floor plans, it also considers the possibility of expansion, adding 10 m$^2$ of space. On the other hand, social housing floor plans exhibit various forms. They differ in shape, including square, rectangular, and 'L' shapes, and the position of the entrance varies, either in the center or on the side.

**Table 5.** Comparison of social housing floor plans and SH specialized floor plans (1 Bed type).

| SH Specialized Floor Plan | Social Housing Floor Plan | | |
|---|---|---|---|
|  |  |  |  |
| SH Green Light (39.8 m$^2$) | Ddim House (36.72 m$^2$) | Banya Ville I (39.88 m$^2$) | Hansol (39.67 m$^2$) |

In the Ddim House floor plan, the entrance at the center leads to a frontal kitchen, with bedrooms, a living room, and a dining area on either side, showcasing a well-separated layout of communal and private spaces. The Banya Ville I floor plan is designed along the entry route through the entrance, with the living room and dining area on one side and bedrooms arranged on the other. It features large balconies on the north and west sides, providing residents with a higher outdoor space experience. The Hansol floor plan is similar to the Banya Ville I floor plan, with bedrooms, living room, and dining area arranged along the entry route through the entrance. It is notably unique, with a relatively large bedroom space, resulting in smaller living room and dining area spaces.

In the 1 Bed type, two types can be classified based on the form of separation between the bedroom and shared spaces. The first type entirely separates the spaces into two sections based on the entrance, providing complete privacy for the bedroom. The second type arranges the shared spaces, such as the living room, kitchen, and dining area, adjacent to the bedroom. This layout offers the advantage of efficiently configuring the overall indoor space.

As shown in Table 6, in the 2 Bed type of SH specialized floor plan, similar to the 1 Bed floor plan, the living room and bedroom are separated in the front area, and a second bedroom is located near the entrance. The universal floor plan forms an 'L' shape, with the kitchen, living room, and dining area in the center and bedrooms on both sides. The advantage lies in the separate placement of bedrooms for residents' privacy. The Hyundai Park floor plan is rectangular, with the entrance leading to the opposite side where the bedrooms, living room, and dining area are situated. It also features separate bedrooms like the Universal, considering residents' privacy. Baek-Yeon Ville's floor plan is square, and based on the entry route from the center, the main spaces are separated. In the 2 Bed social housing floor plans, the most distinct feature is that the main entry route divides the dining area and kitchen, unlike the SH specialized floor plans, which show a different spatial layout.

**Table 6.** Comparison of social housing floor plans and SH specialized floor plans (2 Bed type).

| SH Specialized Floor Plan | Social Housing Floor Plan | | |
|---|---|---|---|
|  |  |  |  |
| SH Green Light (49.91 m$^2$) | Universal (43.74 m$^2$) | Hyundai Park (44.55 m$^2$) | Baek-Yeon Ville (50.00 m$^2$) |

It was challenging to find significant differences when comparing the Studio type of social housing with the SH specialized floor plans. However, as the unit size increases and the bedroom space expands, social housing floor plans exhibit a greater variety of configurations. The SH specialized floor plans are designed for apartment-style buildings in South Korea, with rectangular unit layouts configured with one or two units depending on the size of the front bay. They maintain a similar layout even when expanded. On the other hand, social housing floor plans are primarily focused on small-scale buildings with generally less than 19 units, allowing for more diverse and flexible planning to optimize efficiency on the given site. While it is difficult to find drastically altered floor plan configurations in the SH specialized floor plans designed for apartments, social housing's flexibility allows for various plans to be chosen based on residents' preferences. When comparing SH specialized floor plans and social housing floor plans, the size based on the type showed similarities, but differences were observed in the spatial layout. However, it

was challenging to identify factors where social housing floor plans had significantly more advantages or disadvantages.

## 5. Discussion

First, although social housing targeting young single-person households and newlyweds would consist of small housing units with limited space, the actual classification of social housing types showed a higher percentage (52.6%) of Alcove Studio, 1 Bed, and 2 Bed types compared to the smallest Studio type. A study found that young single households have expressed a desire for larger housing sizes [24]. This indicates that young single households are seeking larger spaces than those provided by the minimum housing standard currently utilized by the government in the supply of rental housing. Therefore, the recently launched social housing project considered the current needs of the younger generation.

Second, this study found regarding the residential space size, Studio units below 30 m$^2$ accounted for the highest percentage (31.7%), and when combined with 1 Bed and 2 Bed units, the total percentage reached 36.7%. On the other hand, units larger than 31 m$^2$, which included Studio, 1 Bed, and 2 Bed types, constituted 53.3% of the total, indicating a higher proportion of units with relatively more spacious living areas. This finding aligns with a study that revealed a demand for separate bedrooms and living spaces and a desire to expand the living area where various activities occur [25]. This suggests residents prefer having distinct, separated spaces even if it results in smaller room sizes. This preference is for private spaces independent from the bedroom [26].

Third, social housing displayed various spatial configurations. These findings show that various layouts are a branding tool for Seoul's social housing that allows residents, particularly the younger generation, to choose their living style according to their preferences [10]. This trend reflects a move towards developing various forms of living spaces that offer flexible and optional features according to the demands of the consumers [26].

## 6. Conclusions

This study analyzed the indoor spaces of social housing in Seoul and compared them with the SH specialized floor plans to analyze the spatial aspects of ongoing social housing projects. Although smaller studio units are expected to be more prevalent in social housing targeting young singles and newlyweds, over half of social housing units are larger Alcove Studio, 1 Bed, or 2 Bed types. Similarly, more than half of the total social housing units were units larger than 31 m$^2$. These findings suggest that most residents prefer separate rooms and units of a reasonable size.

Comparing the floor plans with SH specialized ones, the differences in scale were not significant. Social housing tended to adopt diverse plans that were more flexible and adapted to the specific site conditions, rather than following a standardized SH specialized plan designed for apartment buildings.

Contrary to the expectation of limited budgets and cost constraints resulting in smaller and more compact spatial layouts, the current social housing plans consist mainly of more universally accepted and well-balanced floor plan configurations.

Based on the analysis results, social housing in Seoul primarily caters to young single-person households and newlyweds, with small and efficient living spaces as common characteristics. However, the prevalence of Studio, Alcove Studio, 1 Bed, and 2 Bed types indicates that residents prefer units with more diverse spatial configurations. Social housing offers a variety of spatial layouts compared to SH specialized plans, which allows residents to choose a plan that suits their preferences and site conditions.

Clear guidelines for the interior space of social housing are essential. Currently, there is a lack of clear guidelines for the interior space planning of social housing in Seoul. As a result, the spatial layouts and forms of social housing vary depending on the interpretation and intentions of the designers. It is crucial to conduct more research and incorporate

residents' opinions during the planning process to design the interior and exterior spaces of social housing.

**Author Contributions:** Conceptualization, S.-H.C. and J.H.K.; methodology, S.-H.C. and J.H.K.; software, S.-H.C.; validation, J.H.K.; formal analysis, S.-H.C. and J.H.K.; investigation, S.-H.C.; resources, S.-H.C. and J.H.K.; data curation, S.-H.C.; writing—original draft preparation, S.-H.C.; writing—review and editing, J.H.K.; visualization, S.-H.C.; supervision, J.H.K.; project administration, J.H.K. All authors have read and agreed to the published version of the manuscript.

**Funding:** This research received no external funding.

**Data Availability Statement:** Data sharing is not applicable to this article.

**Conflicts of Interest:** The authors declare no conflict of interest.

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
