# Peer review of "The Spatial Analysis and Enhancement of Social Housing in Seoul"

_buildings, doi:10.3390/buildings13102643_

Round 1
Reviewer 1 Report
The introduction is good in that it presents a clear research problem for the supply of social housing within Seoul, Korea. It does this be adequately comparing Seoul with other densely populated cities around the world.
However, this article’s focus is specifically on the spatial analysis of this social housing, so I think there could be a bit more explanation of this need in the introduction.
‘European countries, which have implemented social housing policies earlier than South Korea, have gone through trial and error to enhance the spatial functions and utilization of social housing to improve citizens' quality of life. This underscores the importance of spatial analysis, given that the spatial functions and utilization of social housing directly impact residents' quality of life.’ – For example it might be useful to expand on this paragraph, highlighting a few of the lessons learned to really underscore the importance of spatial analysis.
The social housing in Seoul section is well put together and succinctly paints a picture of the Seoul Social Housing landscape. It is easy to review the diagram in particular and understand the correlations that the authors are discussing. However, similar to the introduction, if the paper is specifically discussing the ‘spatial analysis’ of this social housing, then I would expect to see more literature or examples introduced that focus strongly on this specific aspect. For example, just a brief overview explaining ‘what is spatial analysis?, what have others written about spatial analysis, or how has it been used, that will be valuable in relation to Seoul. Conversely, what is missing from spatial analysis literature that makes this article a contribution.
Similarly, in this section, I would expect a more detailed outline of exactly what Spatial Analysis is, what is spatial configuration and how are these units examined or analysed? I think there is value in clarifying and defining these terms, so that readers who may not be familiar with this form of analysis, can quickly understand the value of this contribution.
3.2. Spatial Analysis
First, the characteristics of social housing units were analyzed. Second, the spatial configuration of these units was examined. Third, the spatial configurations of social housing in Seoul were compared with public rental housing. In Seoul, public rental housing was provided by SH, a local public enterprise established under the Seoul Metropolitan Government. The goal of the SH is to stabilized housing of Seoul citizens and improve their quality of life through public rental housing [22].
The results section is detailed and clear to follow. The diagram helps highlight the amount of different social housing in Seoul. I think it’s probably worth separating the Comparisons into it’s own section. I was expecting some sort of Discussion heading, but the discussion wasn’t there. Instead, the Comparisons section felt like a discussion, but what within the results section. I would instead expect the Results to demonstrate the information, and then the Discussion to reflect on this information, in regard to your initial research problem.
‘Despite the expectation that social housing targeting young single-person households and newlyweds would consist of small housing units with limited space, the actual classification of social housing types showed a higher percentage (52.6%) of Alcove Studio, 1Bed, and 2Bed types compared to the smallest Studio type.’ – for example this paragraph is within your conclusion, but I would consider this a part of your discussion that could be expanded upon.
I think overall this paper is well put-together and is detailing a nice contribution for understanding the space of Seoul social housing. However, I would like to understand more about the authors reflections on what they found. It is clear from the introduction that Seoul’s Social Housing is lesser than other capital cities, and that properties are expensive relative to other cities. But it would be useful to have some kind of research question or hypothesis presented that highlights what this papers specific contribution to research is.
Perhaps with this more clearly defined, it can be extrapolated in Section 2, compared against other literature, and then further in the discussion, where the results can influence this contribution and compare this with other spatial analysis studies.
English is very good, and overall the paper is easy to follow. There are just a few sections that need review.
For example, the section Methodology is spelled incorrectly. furthermore, in the abstract we would normally say 'that impacts the residents quality of life.' Rather than, impacts residents the quality of life.
Just review the paper once more and try to address these very small discrepancies where present.
Reviewer 2 Report
This is an important paper on the living space of social housing in South Korea, apparently the most expensive city in the OECD hierarchy when it comes to home ownership.
The paper needs extensive proof-reading and editing. Please see attached file for some ideas.

It needs extensive proof-reading and editing for comprehension.
